# DYNAMO-GRASP: DYNAMics-aware Optimization for GRASP Point Detection in Suction Grippers

**Boling Yang\*[1], Soofiyan Atar\*[1], Markus Grotz[1], Byron Boots[1], Joshua R. Smith[1]**
[1] The University of Washington * Authors Contributed Equally

**Abstract:** In this research, we introduce a novel approach to the challenge of suction grasp point detection. Our method, exploiting the strengths of physics-based simulation and data-driven modeling, accounts for object dynamics during the grasping process, markedly enhancing the robot's capability to handle previously unseen objects and scenarios in real-world settings. We benchmark DYNAMO-GRASP against established approaches via comprehensive evaluations in both simulated and real-world environments. DYNAMO-GRASP delivers improved grasping performance with greater consistency in both simulated and real-world settings. Remarkably, in real-world tests with challenging scenarios, our method demonstrates a success rate improvement of up to 48% over SOTA methods. Demonstrating a strong ability to adapt to complex and unexpected object dynamics, our method offers robust generalization to real-world challenges. The results of this research set the stage for more reliable and resilient robotic manipulation in intricate real-world situations. Experiment videos, dataset, model, and code are available at: https://sites.google.com/view/dynamo-grasp.[1]

**Keywords:** Suction Grasping, Manipulation, Deep Learning, Vision

## 1    Introduction

Grasp point detection is essential for successful robotic manipulation, as it requires identifying the optimal location on an object for a robot to securely grasp and manipulate. Rapid and reliable grasping capabilities for a wide range of objects can benefit various applications, such as warehouse and service robots. Suction grasping is a popular grasping modality in real-world settings due to its simplicity and reliability when handling objects with nonporous, flat surfaces compared to parallel-jaw or multi-finger grasping. Existing methods for finding suitable grasping areas for suction grippers typically focus on maximizing suction seal quality and robustness against wrenches, taking into account the object's shape, size, and surface properties [1, 2].

Most existing methods for suction grasping assume a top-down manipulation setting, where objects are initially placed on a stable, flat surface before being grasped, and the robot attempts to grasp the object from above. This is due to the suction cup gripper requiring the robot to apply a specific amount of force to press the suction cup against the object's surface, which causes the cup to deform and create an air seal, resulting in a secure suction grasp. Consequently, an object being grasped needs sufficient and stable support in the direction opposite the robot's pushing. Without such support, the object may move in an unfavorable direction, leading to the suction cup's failure to form the air seal. However, numerous real-world scenarios require a robot to grasp objects without stable support, such as grasping from a container with a side opening or from an unstable pile of objects. In these situations, the objects may exhibit significantly more complex dynamics during the manipulation process due to the displacement caused by the robot's motion and the objects' interactions with one another. State-of-the-art grasp point detection methods for suction grasping could suffer from these complex object-picking scenarios because they do not consider the objects'

---

[1]Our dataset and code are open-sourced at: https://github.com/dynamo-grasp/dynamo-grasp

movement during the manipulation process. This limitation greatly restricts the range of scenarios in which suction grippers can be applied, preventing them from reaching their full potential in real-world manipulation tasks. Fig.1.a. illustrates a real-world manipulation task.

In this work, our goal is to fully exploit the potential of suction grippers by developing a grasp point detection model that not only examines quantitative metrics such as suction quality but, more importantly, considers the object dynamics during the picking process. This paper makes the following contributions: **1. Suction Grasping by Taking Object Movement into Consideration:** We describe the challenge of complex object movement during suction grasping, which no current state-of-the-art method adequately addresses. **2. An Open Source Novel Suction Grasping Simulation:** To address this challenge, we developed a high-performance suction grasping simulation environment using Isaac Gym[3]. This simulation environment models the influence of object dynamics on the success of suction grasps throughout the grasping process. **3. A Dataset and Learned Model:** Utilizing the simulation environment, we generated a dataset that contains more than one million simulated grasps and trained a grasp point detection model that takes into account how the movement of objects and the robot's kinematics impact the success of grasping. **4. Evaluation in a Real-world Warehouse Setting:** We assessed two grasp point detection approaches alongside our model. In both simulated and real-world experiments, our method surpassed the alternatives in terms of accuracy and consistency.

## 2   Related Work

Suction-based robot manipulators have gained widespread popularity in real-world applications. For instance, suction grasping methods are used in manufacturing [4, 5, 6], warehousing [7, 8], underwater manipulation [9, 10], food and fruit manipulation [11, 12, 13, 14], etc. Another major direction where suction grasping has been applied is in the exploration of end-effector modalities [15, 16, 17, 18].

**Analytic Models.** In the realm of conventional suction cup grippers, the effective analysis of grasp quality necessitates the modeling of various cup properties. Given that these suction cups are typically fashioned from elastic materials, such as rubber or silicone, researchers frequently employ spring-mass systems to represent their deformations [1, 2, 19]. Upon establishing a secure grasp on an object using a suction gripper, the suction cup is typically modeled as a rigid entity. The subsequent analysis involves assessing the forces imposed on the object, encompassing those along the surface normal, friction-induced tangential forces and suction-generated pulling forces [20]. Mahler et al. [1] introduced a combined model in DexNet3.0, incorporating both torsional friction and contact moment within a compliant model of the contact ring between the cup and the object. This amalgamated model has demonstrated its efficacy and is employed in

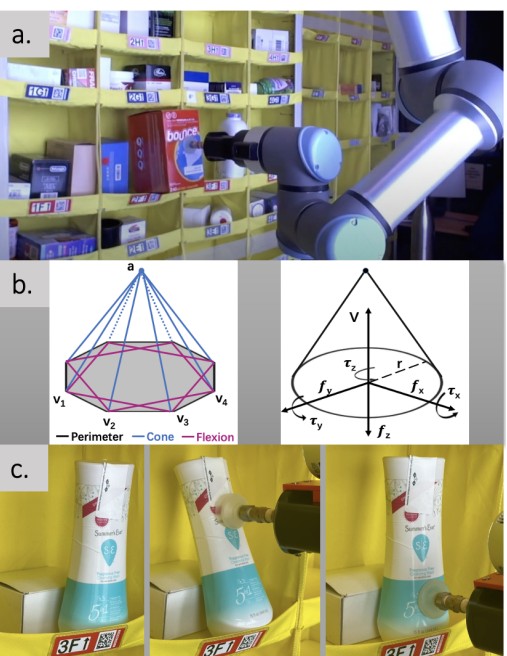

Figure 1: a. Suction grasping for real-world scenarios remains challenging due to limited analysis of object movements. b. SOTA methods only reason for object's surface properties. *Left*: The quasi-static spring model. *Right*: Wrench basis for the suction cup. [1] c. *Left*: A warehouse picking scenario. *Middle*: DexNet failing the grasp due to object toppling. *Right*: An effective grasp point that prevents unfavorable object movements. See the project website for experiment videos.

subsequent works [2, 21]. Additionally, this work adapts the analytic models from DexNet3.0 for the purpose of data annotation.

**Learning Suction Grasps.** Machine learning research in robotics has been actively exploring the selection of optimal grasp points to enhance suction grasping for intricate manipulation tasks [22, 23]. These tasks include novel object picking, object stewing, picking from containers, etc. Existing approaches generate training data through either human expertise [24] or simulations [1, 2, 22, 25]. DexNet3.0 [1], for instance, synthesizes training data and proposes suction grasp points that aid in forming an effective suction seal and ensuring wrench resistance. Several other studies center around clustered scenarios by creating models that take RGB-D input and predict graspable points [2, 25, 24]. Jiang et al. [22] proposed a methodology that simultaneously considers grasping quality and robot reachability for bin-picking tasks. Despite these studies primarily focusing on analyzing surface properties or robot configuration, they largely overlook how the displacement of the object during the picking process might impact the success of the task. Addressing this particular aspect is the main focus of our work.

**Visual Pushing.** This project also shares relevance with the active research area of object displacement modeling during manipulation [26, 27]. Effective non-prehensile manipulation strategies have been successfully applied to enhance grasping operations [28, 29, 30]. Recently, reasoning object translation via visual input has gained huge advances. Transporter and its variants [31, 32] have introduced a data-efficient learning paradigm that links visual inputs to desired robotic actions. Nonetheless, these methods are underpinned by a strong assumption of translational equivariance in visual representation, a condition that is often not met in non-table-top settings. Visual foresight methods [33, 34] have offered a model-based framework that predicts future observations based on a state-action pair. However, these approaches necessitate searching through the action space given a specific task, which can be time-consuming for intricate real-world problems. Other existing studies [35, 36, 37] have examined robotic manipulation from a sideways perspective. However, none of them have explicitly modeled the complex dynamics caused by the interaction between the robot and the objects.

## 3    Problem Statement

Our objective is to identify grasp points on a target object within a container filled with multiple items, using a single-view depth image observation. The identified grasp points should enable a robot to successfully establish a suction grasp, even when the object lacks stable support in the direction opposite to the robot's push. Consistent with previous suction grasp point detection studies [1, 2, 21], a grasp point is defined by a target point $[\mathbf{p}, \mathbf{v}]$. Here, $\mathbf{p} \in \mathbb{R}^3$ represents the center of the contact ring between the suction cup and the object, while $\mathbf{v} \in \mathbb{S}^2$ denotes the gripper's approach direction. The grasp labeling function is defined as $1$ if the grasp successfully forms a suction grasp on the target object, and $0$ if it does not. This section discusses the crucial criteria in order to form a successful suction grasp.

### 3.1    Seal Quality and Wrench Resistance

A suction cup is capable of lifting objects owing to a differential in air pressure. This differential is created across the cup's membrane by a vacuum generator, which pulls the object toward the cup. Ensuring a tight seal between the suction cup and the target object is crucial for successful operation. For sealing evaluation, we follow the highly effective quasi-static spring-based model proposed in DexNet 3.0 [1]. As shown in Fig.1.b., this model uses a combination of three spring systems to represent suction cup deformation. A perimeter springs system is used to assess the deformation between adjacent vertices, namely $v_i$ and $v_{i+1}$. The cone springs system signifies the deformation of the suction cup's physical structure, as determined by the distance between $v_i$ and $a$. Lastly, the flexion springs, which connect vertex $v_i$ to $v_{i+2}$, are employed to resist bending along the surface of the cup.

When the suction cup forms an air seal with the object, the suction gripper should be able to resist wrenches that are caused by gravity or other disturbances. The suction ring contact model proposed in DexNet 3.0 [1] efficiently encapsulates the forces experienced by a suction cup during a grasp. As depicted in Fig.1.b, this model takes into account five forces. The actuated normal force ($f_z$) and

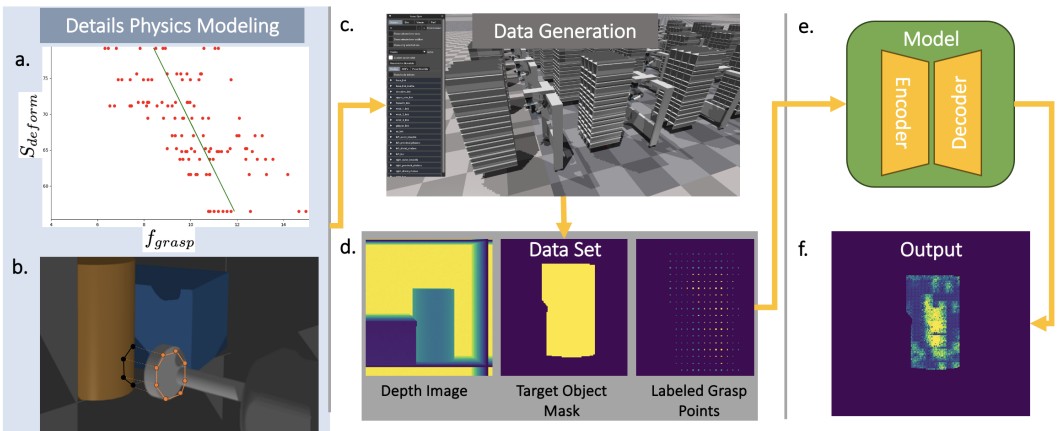

Figure 2: An overview of the proposed pipeline: **a.** We conducted system identification using 19 everyday objects of diverse shapes, weights, volumes, and materials to ascertain the function $F$ discussed in Section 4. **b.** Calculation of deformation score at each simulation time step. **c.&d.** Generating dataset with our simulation environment. **e.&f.** Trained DYNAMO-GRASP model outputs an affordance map highlighting optimal grasp areas.

vacuum force $V$ represent the gripper pressing into the object along the contact axis and air suction pulling the object, respectively. The friction forces $(f_x, f_y)$ and torsional friction $(\tau_z)$ result from the normal force exerted between the suction cup and the object, acting as resistive forces. Lastly, the elastic restoring torques $(\tau_x, \tau_y)$ result from the elastic restoring forces within the suction cup, which apply torque on the object along the boundary of the contact ring.

## 3.2 Object Movement

Most established suction grasping techniques assume little to no movement of the object during the process, which facilitates the deformation of the suction cup, thereby enabling the formation of an air seal for a secure grasp. However, in various practical manipulation scenarios, the target object might not have ample and steady support opposite the robot's push. This lack of support can lead to undesirable shifts in the object's position, preventing the successful creation of the air seal. The situation becomes even more complex when other objects are located near the target, due to the interactions among them. This work addresses these complexities by modeling the movement of objects during the picking process, which enhances the applicability and efficiency of suction-based grippers in real-world manipulation tasks. Assuming that an object's state is denoted by its Cartesian pose and velocity in a workspace, represented as $s = (p, \delta p)$, the states of $i$ objects in a container at time $t$ can be represented as $\mathbf{s_t} = \{s_{t_0}, s_{t_1}, ..., s_{t_i}\}$. At each time step, a robot equipped with a suction gripper performs a pushing action $a_t = (f_t, \mathbf{p}, \mathbf{v})$, applying a force $f_t$ to a specific location, $\mathbf{p}$, on the object's surface in the direction of $\mathbf{v}$. The state transition model $p(\mathbf{s_{t+1}}) = T(\mathbf{s_t}, a_t)$ provides a distribution over the potential movements of the objects during the picking process.

## 4 DYNAMO-GRASP

This section proposes a robot learning pipeline designed to create a grasp point detection model. This model suggests suction grasp points by analyzing combined information regarding object surface properties and object movement during the picking process. We first implemented a new suction grasping simulation environment that accurately simulate suction cup properties and objects' displacement caused by the robot's motion and the objects' interactions with one another. A transformer-based model is trained to take a depth image as input and generates an affordance map over the target object's surface, indicating the likelihood of a successful suction grasp if a robot executes a pushing action along the surface normal across various areas of the object. Please note that our method primarily focuses on analyzing the impact of physical interaction between robots

and objects on the quality of a suction grasp. During execution, we filter out grasp points that offer inadequate air seals and wrench resistance, based on DexNet's output. Fig.2 shows the system architecture.

## 4.1 Simulation Environment and Data Generation

In order to eliminate the need for expensive real robot data collection, we carefully designed a simulation environment that accurately replicates the physical properties of the suction cup, the motion of objects caused by robot grasping, as well as the robot's kinematics during the picking process. We chose to implement our grasping simulation environment based on Isaac Gym, allowing all computations to be accelerated via GPUs. While Isaac Gym lacks important features that emulate detailed suction grasping properties, our environment integrates several custom-implemented functional modules. It provides a pipeline that accurately simulates a suction-picking process by taking into account factors such as suction cup properties, robot kinematic constraints, collisions, control noise, and object dynamics. Additional implementation details can be found in Appendix.A.1

**Modeling Suction Properties.** The majority of popular physics simulations for robotics merely simulate suction grasp through simplistic mechanisms. These mechanisms typically involve directly attaching the object to the robot's end-effector or creating an attracting force between the object and the effector. However, these approaches neglect critical physical details. Specifically, to successfully register a suction grasp, the suction cup must be pushed and deformed to a sufficient extent that the rim of the suction cup attaches to the surface of the object, thereby forming an air seal. Modeling the amount of force required to form an air seal is crucial for this problem. This is because when the target object lacks rigid support, exerting sufficient force directly causes the object's displacement. Understanding the magnitude of the force that the robot exerts on the target object is instrumental in recreating accurate object dynamics. To model the deformation properties of the suction cup, we first adopted the Perimeter Springs in the quasi-static spring system, as discussed in Sec 3.1. Given a grasp point $\mathbf{p}$ on the object's surface and the angle of incident $\mathbf{v}$, this model calculates a suction deformation score $S_{deform} = 1 - \max(r_1, r_2, ..., r_n)$, where $r_i = \min(1, |(l_i' - l_i)/l_i|)$. Here, $l_i$ represents the original length of the perimeter spring linking vertex $v_i$ and $v_{i+1}$, and $l_i'$ is the length after projecting the vertices onto the object's surface. Using real-world data, we then conduct a system identification process to ascertain the function $F$. This function signifies how forcefully the robot needs to press the suction cup to achieve a successful grasp, given a deformation score of a specific grasp point: $F(S_{deform}) \rightarrow f_{grasp}$.

**Simulating Grasping Physics.** *(1) Kinematics:* Our simulation accepts a robot's model as input and controls the robot using an end-effector controller to attempt various suction grasps. This approach enables the simulation to demonstrate how the robot's form factor and kinematic properties impact its grasp. For instance, some grasp points might be physically unattainable for the robot due to its manipulability and reachability constraints or collisions. *(2) Generating Scenarios:* Our experiment primarily focuses on a warehouse lateral picking scenario. During our data generation process, the simulation randomly selects one to three objects from our object set and spawns them into the same container with random positions and orientations. We also implement domain randomization for observation noise, objects' weights, and controller parameters, ensuring the dataset reflects a range of diverse physical properties and robot behaviors. One of the objects in the container is randomly assigned as the target object to be picked. *(3) Sampling Grasp Points:* Given a picking scenario, we sample two sets of candidate grasp points from the visible surface of the target object. The first set is derived from uniform sampling across the entire surface, ensuring that the robot explores diverse picking strategies. The second set contains the grasp points with the highest score returned by DexNet via the Cross-Entropy Method (CEM) sampling strategy, ensuring the robot explores areas that DexNet deems preferable.

**Labeling Data.** After sampling the candidate grasp points, our simulated robot 'physically' executes each candidate $\mathbf{p}$ by performing a sequence of pushing actions $\mathbf{A} = \{a_t\}_{t=0}^T$, where each action $a_t = (f_t, \mathbf{p}, \mathbf{v})$. Here, $\mathbf{v}$ is determined by the surface normal at $\mathbf{p}$. The robot exerts a constant force $f_t = f_c$ if the target object is unstable and moves in response to the gripper's push.

Once the object finds a position with adequate support against the push, $f_t$ gradually increases until the suction cup deforms enough to form an air seal or until the object starts moving again. The simulation of the suction cup's deformation and the precise estimation of suction grasp registration involves a continuous calculation of $S_{deform}$ and $f_{grasp}$ at each timestep. This process considers the suction cup's current position relative to the target object, as shown in Fig.2.b. A force sensor on the end-effector continuously monitors $f_t$, and a grasp is deemed successful if $f_t \geq f_{grasp}$. Any failure to meet this condition, such as collisions or inaccuracies in end-effector positioning due to manipulability or reachability issues, results in the grasp point being marked as unsuccessful. For successful grasps, we incorporated a penalization term $p_{move}$ into the label to penalize unnecessary object movements. Further details are discussed in Appendix.A.1.

## 4.2 Model Training

We employ the suction grasping simulation, as described above, to generate a dataset. This dataset represents a warehouse scenario where a robot equipped with a suction gripper is tasked with extracting a target object from a small container filled with multiple unorganized items. The experimental setting is detailed in Sec.5. This dataset was used to train a model for grasp point selection. The model takes a single-view point cloud of the container's interior, a segmentation mask identifying each object within the container and its boundaries as inputs. It then outputs an affordance map representing the estimated probability of successful grasps at all potential grasp points on the target object. The largest value in the affordance map indicates the optimal grasp point, $(\mathbf{p}^*, \mathbf{v}^*)$, for the given scenario. As shown in Fig.2.e, our model employs an auto-encoder architecture, integrating a transformer encoder and a deconvolutional decoder. As previously mentioned, our data generation process is designed to capture the inherent variabilities of complex real-world robotic suction grasping tasks. These include variations stemming from the physical properties of different objects, robot constraints, and stochasticity in the controller, among others. Empirically, we discovered that the following loss function can effectively mitigate the adverse effects of the high aleatoric uncertainty in our dataset during training: $\mathcal{L}_{Y_{max}} = \frac{1}{N} \sum_{i=1}^{N} (y_i - \hat{y}_i)^2, \forall y_i \in Y_{max}$. Here, $Y_{max}$ represents a subset of samples containing the $n$ highest-scored grasp points on an object. More details are discussed in Appendix.A.2.

## 5   Experiment

Our experiment focuses on robotic suction grasping for industrial warehouse shelves [38]. Fig.1 depicts the robot setup and the industrial shelving unit which is packed with objects. The opening of these shelving units is located on the side, which makes suction grasping significantly more challenging compared to top-down manipulation scenarios, as the robot's movements can trigger a series of object displacements, leading to objects being shifted or even toppled over. Consequently, this scenario serves as an excellent evaluation environment for our work. Our system setup is as follows. Throughout our evaluation, we employed a

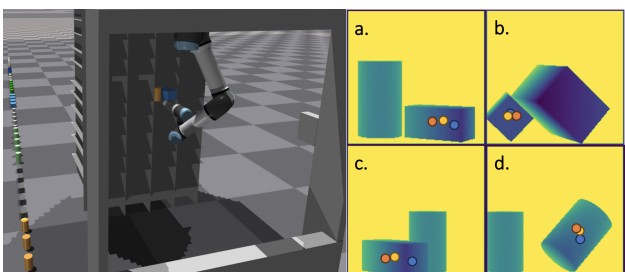

Figure 3: **Left:** The simulation environment for data generation and experiments. The simulated objects with different weights, sizes, and shapes are displayed on the left side of the robot. **Right:** In Section 5.1, challenging test cases are presented where only DYNAMO-GRASP was successful in grasping the target object. The orange, blue, and yellow points indicate the grasp points proposed by DYNAMO-GRASP, DexNet, and the Centroid method, respectively.

Universal Robots UR16e robot equipped with a Robotiq EPick suction gripper and an Intel Realsense L515 camera mounted on its wrist. A large variety of objects with different shapes, dimensions, and physical properties were used in our experiment, details can be found in Appendix.A.1.

|                    | Dyn(Full) | Dyn w/ MSR | Dyn w/o PEN | Dex    | Cen    |
| ------------------ | --------- | ---------- | ----------- | ------ | ------ |
| Total Success Rate | 88.05%    | 86.75%     | 82.93%      | 81.12% | 78.78% |
| Success Std        | 0.30      | 0.32       | 0.36        | 0.36   | 0.40   |

Table 1: The first row of the table displays the grasping success rate for each method, calculated from all 1300 picks. The second row provides the standard deviation of the success rate for each method across various scenarios. The first three columns of the table present an ablation comparison for our DYNAMO-GRASP (*DYN*) method, while *Dex* and *Cen* represent the DexNet and Centroid methods, respectively.

In our experiment, we focus on evaluating three methods: 1. our method DYNAMO-GRASP (Dyn), 2. DexNet3.0 (Dex), and 3. the Centroid method (Cen). DexNet3.0 is a SOTA suction-picking technique, serving as a strong baseline. Meanwhile, the Centroid method, a straightforward approach involving suctioning on the object's centroid, has proven effective in similar tasks at the Amazon Robotics Challenge [37, 39].

## 5.1 Large-scale, Diverse Scenario Assessment, and Ablation Test

To comprehensively assess the performance and robustness of various methods for the suction grasping challenge, we generated 260 diverse picking scenarios. We use the same simulation environment as we used to generate our training dataset. Each of the three methods was tested with five suction grasps per scenario in simulation, resulting in 1300 simulated suction grasps for each method's evaluation. The scenarios were generated by sampling from a distribution that incorporates even greater randomness in object orientation than the dataset used for model training. These scenarios incorporate a wide range of object configurations, leading to potentially complex object movements during picking.

Comparing the first, fourth, and fifth columns of Table.1, it is evident that our method exhibits a marked improvement over both DexNet and the Centroid method in terms of overall success rate and consistent performance across various scenarios. **Our method achieved the highest success rate of 88.05% and exhibited the least variance in success across different scenarios.** The first, second, and third columns of Table.1 presents an ablation test that illustrates the contributions of various components in our learning pipeline to the effective training of our model. *Dyn(Full)* is our final model, *Dyn w/ MSR* represents a model trained with standard MSR loss instead of the $\mathcal{L}_{Y_{max}}$ described in Sec.4.2, and *Dyn w/o PEN* further remove the use of penalization term $p_{move}$ in the labeling process.

## 5.2 Real-world Evaluation

To assess real-world efficacy, we executed 375 real-world suction grasps to evaluate the various methods. In this experiment, we curated three sets of scenarios: the *Common set*, *Challenging set*, and *Adversarial set*, each embodying a distinct level or type of challenge for suction grasping. The statistic of all experimental trials and their comparison to the simulated trials are detailed in Table.2, 3, and 4 in Appendix.A.3.1.

**The Common Set:** In this experiment, we sampled ten scenarios from the 260 randomly generated ones as detailed in Sec.5.1. We then recreated these scenarios in the real world using objects with dimensions similar to those in the simulations. Each method was used to perform five grasps on each of these scenarios. This evaluation set captures the typical challenges of most picking tasks in this specific warehouse environment. As shown in Fig.4, **our model demonstrates an advantage with a total success rate of 94%, averaging 4.7 successful grasps out of five attempts and a standard deviation of 0.67**. In contrast, both DexNet and the Centroid method average 4.2 successful grasps out of five attempts. Their higher standard deviations, 0.92 and 1.03 respectively, point to less consistent performance.

**The Challenging Set and Adversarial Set:** We are particularly interested in the more challenging cases. Consequently, we devised two sets of scenarios in the real world to further test the capabilities of the three methods. The *Challenging Set* comprises five scenarios from the 260 scenarios

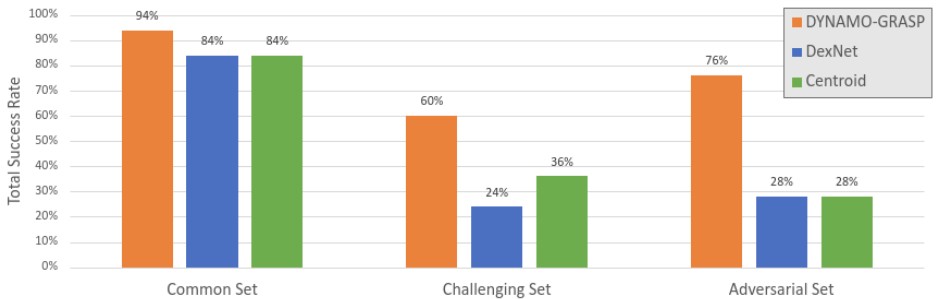

Figure 4: Comparison of the total success rates of different methods underscores their real-world performance on the three evaluation sets described Sec.5.2. The total success rate is computed by dividing the number of successful grasps by the total number of attempts within an evaluation set.

described in Sec.5.1. These scenarios exhibit the lowest combined success rate for the three methods in simulation, representing the most challenging situations our simulation generated without human bias. In contrast, the *Adversarial Set* comprises five scenarios designed by a human operator, specifically tailored to challenge these grippers. The objects featured in this set are everyday items that were not included during the training phase. As depicted in Fig.4 and Table.3,4 in Appendix.A.3.1, **DYNAMO-GRASP markedly outperforms the two baseline methods in both total success rate and performance consistency in more challenging scenarios.** On the challenging set, our method achieved a success rate of 60%, whereas, on the adversarial set, it reached 76%. In stark contrast, DexNet and the Centroid method's success rates are 24% and 36% for the challenging set, with both achieving 28% on the adversarial set. Furthermore, DYNAMO-GRASP consistently executed more than four successful grasps out of five attempts in over half of the scenarios in both sets. Meanwhile, the other two methods faltered, rarely managing even three successful grasps in any scenario within these evaluation sets.

**Qualitative Analysis.** The Fig.7 in Appendix.A.3.1 depicts the grasp points chosen by various methods and indicates the success of each attempt during the adversarial evaluation. The figure offers insights into the areas chosen by each method for grasping and sheds light on which areas are more likely to lead to successful grasps. For example, in the first scenario, a tall bottle is partially propped up by a box in the back. The test checks the grasp method's awareness of potential object toppling. DYNAMO-GRASP chose the bottle's lower part, ensuring the box supported the pick. Some grasp points chosen by the other two methods were higher up on the bottle leading to toppling movements. Similarly, in scenarios two, four, and five, **DYNAMO-GRASP tends to select grasp points from regions that are overlooked by the other methods, resulting in more successful grasps in these scenarios.**

## 6 Conclusion and Limitation

This paper discusses the challenge of complex object movement during suction grasping, which no current state-of-the-art method adequately addresses. We introduced DYNAMO-GRASP, a dynamic-aware grasp point detection method that selects grasp points by factoring in the impact of object movement on the success of suction grasping. DYNAMO-GRASP delivers improved grasping performance with greater consistency in both simulated and real-world settings. Notably, in real-world experiments involving challenging scenarios, our method exhibits an improvement of up to 48% in success rate compared to alternative methods. **Limitations and future work:** Firstly, the dataset used in our simulation environment primarily includes objects with relatively simple geometric shapes. This aspect could limit the efficacy of our method when dealing with objects of uncommon or complex shapes. Similarly, our real-world experiments primarily involved simple geometric objects, such as boxes and bottles. In future research, there's potential to develop effective heuristics that combine information from both DYNAMO-GRASP and DexNet. While our method emphasizes modeling object movement, DexNet primarily targets suction quality based on object surface geometry. Integrating the strengths of both methods could lead to enhanced performance in specific applications.

**Acknowledgments**

This research is funded by the UW + Amazon Science Hub as part of the project titled, "Robotic Manipulation in Densely Packed Containers." We would like to thank Dr. Michael Wolf from Amazon for valuable discussions. We would like to thank Yi Li for many constructive inputs to the machine learning aspect of this paper. We thank all members of the Amazon manipulation project who generously lent us their computing resources and robot time.

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

# A Appendix

## A.1 Simulation Details

A simulation that accurately replicates the targeted robotic tasks can significantly enhance the efficiency of various machine learning algorithms in learning these tasks[40, 41, 42, 43]. Most physical simulators, such as Mujoco[44], PyBullet[45], and IsaacGym[3], which excel at simulating the physical properties of object motion, lack the functionality to simulate the characteristics of suction cups during suction grasping. Our development efforts focus on utilizing the sensing and physical information in IsaacGym to create more realistic suction-picking properties.

**System Identification:** Our system identification process aims to accurately model the force required by the robot to deform the suction cup. This ensures the rim of the cup adheres to the object's surface, forming an air seal. We chose 18 everyday objects with varied surface geometric characteristics, aiming to cover a broad spectrum of deformation scores. For each object, we executed ten suction grasps using our UR16 robot. To minimize measurement noise, the objects were held firmly to limit movement during the grasping process. The force required for the suction gripper to achieve a suction seal was detected by a sudden decrease in suction airflow and the force torque sensor located on the robot's wrist. We observed that the characteristics of our suction cup differ significantly between nearly flat object surfaces and those that are more curved or intricate. Consequently, we chose to represent the function $F(S_{deform}) \rightarrow f_{grasp}$ using a hybrid linear function:

$$F(S_{deform}) = \begin{cases} 7.66 - 0.06 * S_{deform} & \text{if } S_{deform} \leq 80 \\ 22.2 - 0.18 * S_{deform} & \text{otherwise} \end{cases}$$

Typically, the size and firmness of a suction cup influence its working range for objects of varying sizes and weights. However, this doesn't profoundly alter the nature of this grasping problem. For instance, when using a small suction cup to manipulate lighter, smaller objects, these objects typically have less friction with the container and reduced inertia, making them more prone to toppling. However, even though we anticipate a certain degree of generalization to unseen suction cups, we recommend carrying out the system identification process to achieve optimal performance.

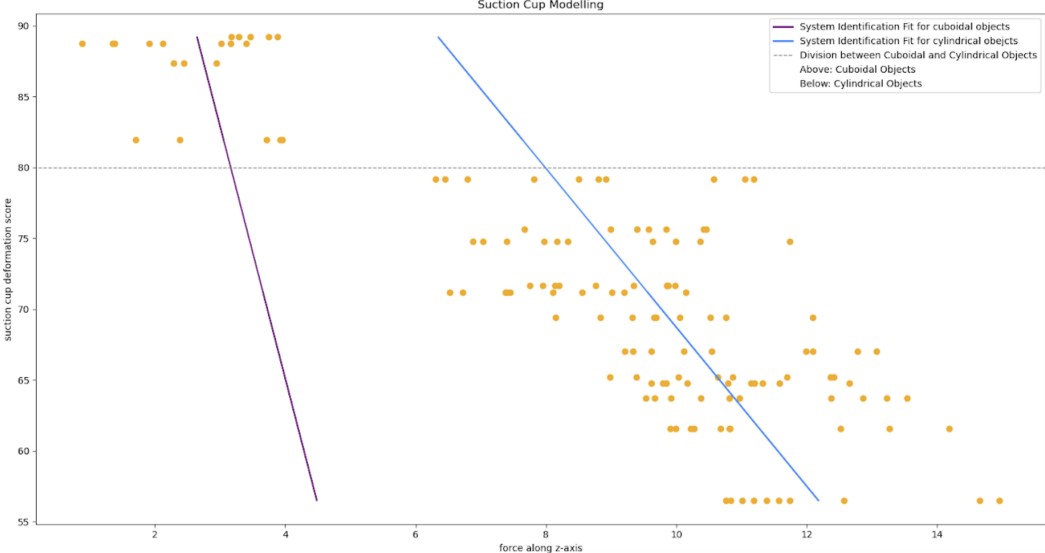

Figure 5: Force exerted on an object as a function of the suction deformation score. Solid lines represent system identification fits for cylindrical (blue-colored line) and cuboidal (violet-colored line) objects. The dotted line demarcates the distribution of data points between the two object types.

**Domain Randomization:** We performed domain randomization to vary object weights, where each of the ten chosen objects had their original weights varied by $-5\,\text{g}$, $-10\,\text{g}$, $5\,\text{g}$, and $10\,\text{g}$,

leading to five weight versions per object. These objects were cylindrical or cuboidal with varying dimensions, inertia, and weights. The cylindrical ones had radii from $26\,\mathrm{mm}$ to $46.3\,\mathrm{mm}$, heights between $95\,\mathrm{mm}$ and $155\,\mathrm{mm}$, and weights ranging from $118\,\mathrm{g}$ to $602\,\mathrm{g}$. In contrast, the cuboidal objects had lengths from $112\,\mathrm{mm}$ to $165\,\mathrm{mm}$, breadths between $55\,\mathrm{mm}$ and $102\,\mathrm{mm}$, and widths from $55\,\mathrm{mm}$ to $95\,\mathrm{mm}$. Notably, with every weight change, the inertia properties were appropriately modified. When placing objects in the simulator, their orientations on all three axes were uniformly picked from $-180°$ to $180°$. Although their initial placements followed predefined bin coordinates, potential collisions might displace some objects. As a preventive measure, we ensured that each object remained within the bin limits and verified the stability of each object setup by spawning it thrice and monitoring its movement at each simulation step for minimal displacement until the suction gripper gets in contact with the target object. Lastly, to closely mimic our physical robot setup with the Intel RealSense L515 camera, we added Gaussian noise (mean: $0\,\mathrm{mm}$, standard deviation: $0.9\,\mathrm{mm}$) to the depth images.

**Labeling:** For the label for each configuration, each grasp point score serves as an indicator of grasp success. A grasp point that fails to achieve a secure suction grip is assigned a definitive zero score. Additionally, the label is designated as a 'failure' if the robotic arm does not align and picks the object at the computed angle of incidence derived from the surface normals, ensuring the grasp adheres to the pre-calculated optimal orientation. Another critical constraint is that the arm must avoid unintended contact with any other object before establishing contact with the target, as such collisions can compromise the grasp's integrity and lead to potential inaccuracies or damage. On the other hand, successful grasp points are scored using the equation $s = 1 - p_{move}$, where,

$$p_{move} = \max(0, \min(obj\_movement, 0.3))$$

$$obj\_movement = \sum_{t=0}^{T-1} \left( ||\mathrm{tran}_{t+1} - \mathrm{trans}_t|| + (1 - |\mathrm{quat}_{t+1} \cdot \mathrm{quat}_t|) \right)$$

$p_{\mathrm{move}}$ is a penalization term that discourages unnecessary movement of the target object. $obj\_movement$ calculates the total movement of the target object during the picking process. The picking horizon $T$ is discretized by a fixed interval, and $t$ represents a time step within $T$. tran and quat represent the translation and orientation of the target object at a given time step, respectively.

**Dataset:** We implemented specific data augmentation techniques on our dataset to enhance our model's resilience against variances in real-world scenarios. We added Gaussian noise to the point cloud data and flipped the input data along with their corresponding labels, strengthening the model's ability to recognize various object orientations and thereby improving its generalization capabilities. These augmentation strategies significantly expanded the diversity of our training dataset, ensuring the model's proficiency in managing diverse input perturbations. The complete dataset, including labels and augmented inputs, consisted of around 12000 configurations, including augmentations, enhancing the dataset's diversity and depth, which occupy approximately 10 GB of storage space.

## A.2 Learning Details

**Model Architecture:** Our model employs a variation of the Vision Transformer (ViT), adopting the architecture from Beyer et al.[46]. We chose ViT because it represents a state-of-the-art architecture widely used in vision classification tasks. Utilizing this architecture demonstrates that a standard network, when trained with our dataset, effectively addresses the challenge of suction grasping in complex object clusters. This is achieved without the need for custom modifications to the model architecture.

**Loss Fuction:** We initially experimented with both the standard MSE loss and Cross-entropy loss but observed only mediocre performance from the model. As highlighted in Section 4, the domain randomization process introduced significant stochasticity to our dataset. Empirically, we found the presented loss function to be more effective in this specific context. The use of $Y_{max}$ is a simple technique designed to mitigate the adverse effects of high aleatoric uncertainty present in the training data. It updates the model by only taking into account the grasp points that the model deems to have

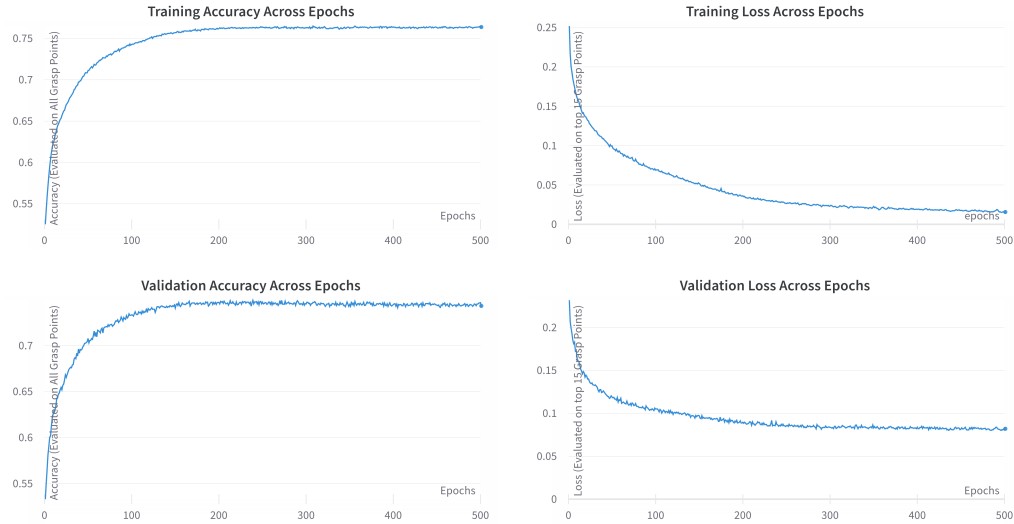

Figure 6: Training and validation metrics over epochs: The top row displays the metrics related to training, with the left graph showing the training accuracy (calculated using all grasp points) and the right graph presenting the training loss (determined with 15 highest-scored grasp points). The bottom row focuses on validation metrics, with the left graph illustrating the validation accuracy (using all grasp points) and the right graph depicting the validation loss (using the 15 highest-scored grasp points)

a high confidence of success. This approach is aimed at penalizing false positive predictions made with high confidence or encouraging true positive predictions made with high confidence while disregarding low confidence labels, which usually arise due to data noise. We discovered that this loss function led to improved prediction accuracy and produced a smoother affordance map.

**Hyperparameters:** We trained our ViT model using the Adam optimizer with a learning rate of $5e^{-5}$ and a batch size of 128 images. The model converged in about 500 epochs, and the training was conducted on an NVIDIA RTX 3090. Our ViT model consists of eight heads, each with a dimension of 64. Consequently, we set Q, K, and V to 128, 257, and 1536, respectively. The model accepts a $4 \times 256 \times 256$ tensor as input. The first, second, and third channels represent the x, y, and z values of the cropped point cloud observation for the container. The fourth channel provides a segmentation mask that localizes the target object. The model produces a $256 \times 256$ affordance map. Each pixel in this map provides a score ranging from $0$ to $1$. A higher score indicates a more favorable grasp point for achieving a successful suction grasp.

**Segmentation Mask:** Within the Isaac GYM simulator, we adhere to ground truth segmentation masks. For real-robot experiments, we used a specialized method, called "STOW" [47], that combines VITA [48] and the Mask2Former [49], which is tailored for joint unseen object instance segmentation and tracking. The method uses transformer-based architectures and dynamic tracking anchors to handle real-world visuals characterized by dense clustering and substantial intra-frame object displacements.

**Grasp Point Selection:** After getting the affordance map, we first identify the pixels that represent the target object in the map using the segmentation mask. Subsequently, we use the DBSCAN algorithm [50] to cluster regions displaying high-affinity scores exceeding 0.9. During this clustering phase, each cluster encompasses a minimum of five pixels. The final grasp point is defined by the centroid of the cluster with the highest average affinity score.

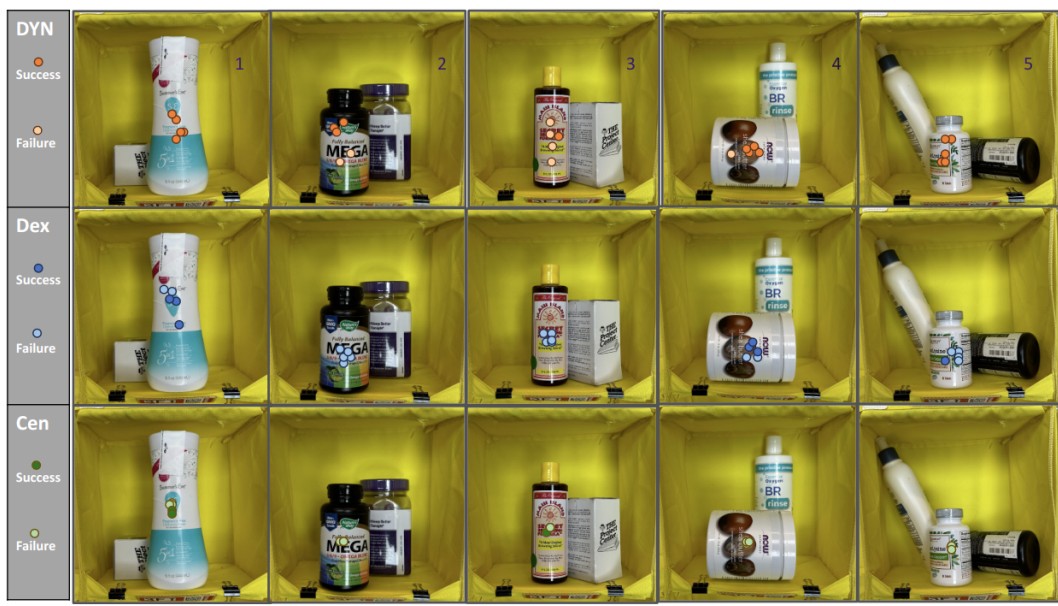

Figure 7: Real-world adversarial evaluation with five grasp points for each configuration: DYNAMO GRASP (our method), DexNet, and Centroid. The color-coded points represent the suggested grasp points success and failure from various algorithms. The successfully identified grasp points are marked by the color along the label "success" and "failure".

## A.3 Experiment Details:

In the real-world environment setup, distinct from the simulator approach, objects were first stowed into a designated bin. Following this, a segmentation algorithm was employed to generate a mask delineating each object. The user then selects the target object based on its unique value in the grayscale mask image, referred to as the 'target object id'. With the object identified, the next phase involves running the inference of a user-provided algorithm to determine the optimal strategy for picking the selected object. The entire operation is orchestrated through a state machine, ensuring a seamless transition between stages. Each state is connected sequentially. In evaluating success and failure across various methods, a grasp point is deemed unsuccessful if motion planning fails consecutively on two occasions. Additionally, if the system does not create a suction with the object, it is also considered a failure. A successful grasp is solely determined by the creation of a good suction with the target object.

### A.3.1 Extra Experimental Result:

**The Sim2Real Gap:** Our DYNAMO-GRASP model was exclusively trained using simulated data. In most of our experiments, this model exhibited outstanding real-world performance without requiring tuning using real-world data. This indicates the model's strong ability to generalize in real-world conditions, showcasing a minimal sim2real gap. To delve deeper into our pipeline's constraints, we pinpointed situations where simulation deemed the target object "impossible" to pick up. In these instances, all three picking techniques registered a 0% success rate in simulation. Importantly, these situations are infrequent, accounting for just around 3% of the 260 simulated test scenarios. We then mirrored these situations in an actual warehouse environment and ran a real robot experiment as delineated in Sec.5.2. Despite the struggles faced by all three methods to secure high success rates (DYN: 16%, Dex: 8%, Cen: 40%), the real-world challenges weren't as formidable as projected by the simulation. However, this did highlight a sim2real gap in these rare scenarios. Our observations also revealed that, in cases where our simulation wasn't entirely accurate, the centroid method surpassed the performance of learning-based approaches. This observation emphasizes the value of refining learning-based models using actual world data.

| Common Set | | | | | | |
|---|---|---|---|---|---|---|
| | Real world experiments | | | sim experiments | | |
| | DYN | Dex | Cen | DYN | Dex | Cen |
| Scenario 1 | 3 | 2 | 3 | 5 | 5 | 5 |
| Scenario 2 | 5 | 4 | 5 | 5 | 5 | 5 |
| Scenario 3 | 5 | 4 | 5 | 5 | 5 | 5 |
| Scenario 4 | 5 | 5 | 5 | 5 | 5 | 5 |
| Scenario 5 | 5 | 5 | 5 | 5 | 5 | 5 |
| Scenario 6 | 5 | 5 | 5 | 5 | 5 | 5 |
| Scenario 7 | 4 | 4 | 4 | 5 | 5 | 5 |
| Scenario 8 | 5 | 4 | 4 | 0 | 5 | 5 |
| Scenario 9 | 5 | 4 | 2 | 5 | 0 | 5 |
| Scenario 10 | 5 | 5 | 4 | 5 | 5 | 5 |
| Avg. Success Grasps | 4.7 | 4.2 | 4.2 | 4.5 | 4.5 | 5 |
| Std. Dev. | 0.675 | 0.919 | 1.033 | 1.581 | 1.581 | 0 |
| Total Success Rate | 94% | 84% | 84% | 90% | 90% | 100% |

Table 2: Comparative evaluation of grasp success rates in common scenarios for three methodologies: DYNAMO-GRASP (DYN), DexNet (Dex), and Centroid (Cen). The table enumerates the average success rates, standard deviations, and total success rates for each method.

| Challenging Set | | | | | | |
|---|---|---|---|---|---|---|
| | Real world experiments | | | sim experiments | | |
| | DYN | Dex | Cen | DYN | Dex | Cen |
| Scenario 1 | 4 | 1 | 2 | 5 | 0 | 0 |
| Scenario 2 | 2 | 2 | 3 | 0 | 3 | 3 |
| Scenario 3 | 4 | 1 | 0 | 5 | 0 | 0 |
| Scenario 4 | 0 | 1 | 3 | 0 | 0 | 2 |
| Scenario 5 | 5 | 1 | 1 | 5 | 0 | 0 |
| Avg. Success Grasps | 3 | 1.2 | 1.8 | 3 | 0.6 | 1 |
| Std. Dev. | 2 | 0.447 | 1.304 | 2.739 | 1.342 | 1.414 |
| Total Success Rate | 60% | 24% | 36% | 60% | 12% | 20% |

Table 3: Comparative evaluation of grasp success rates in challenging scenarios for three methodologies: DYNAMO-GRASP (DYN), DexNet (Dex), and Centroid (Cen). The table enumerates the average success rates, standard deviations, and total success rates for each method.

| Adversarial Set | | | |
|---|---|---|---|
| | Real world experiments | | |
| | DYN | Dex | Cen |
| Scenario 1 | 5 | 3 | 2 |
| Scenario 2 | 3 | 0 | 0 |
| Scenario 3 | 1 | 0 | 1 |
| Scenario 4 | 5 | 3 | 1 |
| Scenario 5 | 5 | 1 | 3 |
| Avg. Success Grasps | 3.8 | 1.4 | 1.4 |
| Std. Dev. | 1.789 | 1.517 | 1.14 |
| Total Success Rate | 76% | 28% | 28% |

Table 4: Comparative evaluation of grasp success rates in adversarial scenarios for three methodologies: DYNAMO-GRASP (DYN), DexNet (Dex), and Centroid (Cen). The table enumerates the average success rates, standard deviations, and total success rates for each method.

