# OpenReview forum: "DYNAMO-GRASP: DYNAMics-aware Optimization for GRASP Point Detection in Suction Grippers"
_robot-learning.org/CoRL/2023/Conference — CoRL 2023 Poster_

### Official Review · Reviewer_gTny · 2023-06-28

**Confidence:** 5
**Originality:** Good
**Technical Quality:** Good
**Clarity Of Presentation:** Good
**Impact:** 3

**Recommendation:**

Weak Accept: I recommend accepting the paper, but will not argue for my recommendation if the majority of other reviewers have a different opinion.

**Review:**

This paper tackles a relevant and challenging problem in robotic manipulation and grasping: how to generate stable suction grasps on objects without support in the grasp approach direction. I agree with the authors that previous methods have primarily studied grasping from the top-down direction, which is a much more forgiving setting than grasping from the side.

What I learned from the paper is that approaching the object from a direction that displaces it as little as possible is probably a good idea when grasping from the side. Another thing I take with me is that the community should look more into grasping objects in confined spaces that require non-top-down grasp directions.

Originality:

The paper addresses the well-known problem of generating suction cup grasps. However, it is original because it focuses on confined spaces, a setup that has yet to be vastly researched. The simulation approach is also novel.

Quality

The quality of the experiments could be better. For instance, I think the experimental evaluation of DexNet 3.0 is unfair, as the dataset used for training that network is on top-down grasps in non-confined spaces. On the same line, the dataset used to train DYNAMO-GRASP only contains side grasps which are also the setting for the experiments, while DexNet 3.0 is not trained on such grasps. There also needs to be a discussion about how the experimental evaluation might be affected by the stiffness and sizes of the suction cup gripper.

Clarity

The paper is well-written and structured. There are a few typos that I point out below.

- Section 5.1 has a few contractions (we've, it's, etc.) Please rewrite these into the non-contraction form.
- On line 195, you refer to $\mathbf{v}$ as the angle of incident, but $\mathbf{v}$ is actually a vector representing the approach direction.
- On line 125, the sentence ends with a colon, indicating that what follows is a list. However, I do not see such a list. Please take a look at that sentence.

Significance

In terms of significance, the paper highlights the difficulty of grasping objects sideways from a confined space with suction cup grippers. The proposed method seems to be better at this problem than other state-of-the-art. However, as mentioned earlier, a more thorough experimental evaluation should be conducted to determine if that is the case.

Relevance

Based on the paper's content, it does not address a robotic learning problem but rather a robotic grasping problem. For instance, the proposed neural network is similar very similar to previous work on suction grasping [1, 2, 3].

Limitations

The limitation section highlights limitations regarding the relatively homogeneous dataset, which is excellent. However, it should also highlight the low number of real-world experiments and that the baseline was not trained on the same dataset as their method.

Strengths:

- The work tackles a challenging and relevant problem in suction cup grasping.
-  A novel, more realistic approach to simulating suction cup grasps.
- The use of the centroid grasping baseline is an excellent complement and demonstrates how good a simple but non-learning-based method can do.

Weaknesses:
- The experimental evaluation of DexNet 3.0 is unfair, as the dataset used for training that network is on top-down grasps in non-confined spaces.
- The dataset used to train DYNAMO-GRASP only contains side grasps which are also the setting for the experiments, while DexNet 3.0 is not trained on such grasps
- There is no discussion on how the experimental evaluation might be affected by the stiffness and sizes of the suction cup gripper. The gripper used in this paper is considerably larger than the gripper used in the original DexNet 3.0 paper. From intuition, I think a smaller gripper needs less force to create a tight seal, which might be why DexNet 3.0 proposes grasps higher up on the objects.
- The network architecture needs to be better described. I would not be able to re-implement it based on the information in the paper.

[1] Cao, Hanwen, et al. "Suctionnet-1billion: A large-scale benchmark for suction grasping." _IEEE Robotics and Automation Letters_ 6.4 (2021): 8718-8725.
[2] Wong, Alexander, et al. "Fast GraspNeXt: A Fast Self-Attention Neural Network Architecture for Multi-task Learning in Computer Vision Tasks for Robotic Grasping on the Edge." _Proceedings of the IEEE/CVF Conference on Computer Vision and Pattern Recognition_. 2023.
[3] Zeng, Andy, et al. "Robotic pick-and-place of novel objects in clutter with multi-affordance grasping and cross-domain image matching." _The International Journal of Robotics Research_ 41.7 (2022): 690-705.

**Quality Of The Limitations Section:**

Additional details required

**Questions For Rebuttal:**

1. It seems that the statement on lines 166-168 contradicts what is written on lines 168-170. As I understood, your model outputs the grasps with the highest likelihood of succeeding (having a high grasp quality) rather than the grasp that displaces the object the least.
2. What does $f_{grasp}$ stand for on line 201?
3. On line 196, is $l_i^{'}$ the geodesic distance between the vertices on the object or the Euclidean distance? If it is the Euclidean distance, can $l_i^{'}$ ever be larger than $l_i$? If not, you could change the order of $l_i^{'}$ and $l_i$ in $r_i$ and eliminate the absolute value.
4. Could you also do the DYNAMO-DexNet experiment in Section 5.1? It should not take long to complete as it is in simulation.
5. In Figures 3 b and d, the centroid method should work better than or at least as well as DYNAMO-GRASP. Why did DYNAMO-GRASP work in those cases but not the centroid method?
6. Is the simulation environment completely deterministic? If not, please repeat the experiments in Section 5.1 over a few (maybe 5) different seeds.
7. In the real-world experiment, only cylinder-type objects are picked. Such objects should be favorable for DYNAMO-Grasp as the suction cup gripper deforms upon contact. For a fair comparison, also do some real-world experiments with box-type objects.
8. As far as I understood, DexNet 3.0 is not retrained on your dataset. If that is the case, is it fair to compare DYNAMO-GRASP to DexNet 3.0 because the dataset DexNet 3.0 is trained on might not contain that many cylinder-type objects, which the proposed dataset is full of? Maybe it is necessary to compare the DexNet 3.0 dataset and the proposed dataset and check if the objects have similar shapes. If there is a vast discrepancy, DexNet 3.0 should be retrained on an object set with a larger proportion of cylinder-type objects.

**Robotics Focus:**

Sufficient demonstration on hardware

**Summary Of Paper:**

This paper proposes DYNAMO-GRASP, a neural network that proposes suction grasps points on objects without support in the grasp approach direction, such as objects on shelves. The actual neural network is a transformer that inputs a depth image and outputs a pixel-wise affordance map that tells how likely a suction grasp is at that specific pixel. A novel, more realistic suction cup simulation setup is introduced to gather training data. The experimental evaluation on grasping objects from shelves demonstrates that DYNAMO-GRASP achieves a much higher grasp success rate than DexNet 3.0, the current state-of-the-art data-driven suction cup baseline.


**Summary Of Recommendation:**

The proposed DYNAMO-GRASP is a novel neural network that predicts dense pixel-wise suction grasps on objects that do not have any support from the environment in the grasp approach direction, such as objects on shelves. The results are good (better grasp success rates than the baseline), but the methodological contributions with respect to DexNet 3.0 are limited. Therefore, this reviewer recommends a weak reject rating. To improve the paper, the authors should do a more extensive experimental evaluation that includes more real-world grasps and the performance of DexNet 3.0 retrained on the new dataset.

---

### Official Review · Reviewer_dsfY · 2023-07-06

**Confidence:** 4
**Originality:** Good
**Technical Quality:** Very Good
**Clarity Of Presentation:** Very Good
**Impact:** 3

**Recommendation:**

Weak Accept: I recommend accepting the paper, but will not argue for my recommendation if the majority of other reviewers have a different opinion.

**Review:**

While the problem tacked by this work is interesting and well-grounded in real world robotic applications, the execution of this paper needs improvement. Please refer to “Questions For Rebuttal” for more specific remarks.

The most significant weakness of the paper is the lack of evaluation. The most extensive evaluation presented is in simulation, which is solely based on a metric that the proposed method is designed around. Given the presumed difficulty in simulating a deformable suction cup, the simulated scenarios should be reproduced in the real world. The real-world experiments are very limited and could arguably be cherry picked. All scenarios were designed by the authors, and only scenario 4 has more than one execution.

Without empirical support, the merits of this paper are limited. From what I can tell, most of the novelty of this paper lies at the system level. Each of the components in the method is adapted from literature. The deformation model for the suction cup is adapted from DexNet 3.0. Predicting contact points with various versions of autoencoder has been done in suction cup [1], parallel jaw gripper [2], and dexterous hand [3]. However, the provided evaluation does not sufficiently illustrate the benefit of the proposed pipeline.

Presentation of this paper needs improvement. Key details were left out, such as the types of objects used, the method used for fitting function F, the specific autoencoder architecture, etc. There is no supplementary material and the paper website does not clarify the information.

Strengths:

1.	Well-motivated problem with clear relevance to robotics

Weaknesses:

1.	Lack of empirical evaluation, especially real-world results

2.	Missing details in the presentation

[1] Jiang, P., Oaki, J., Ishihara, Y., Ooga, J., Han, H., Sugahara, A., ... & Ogawa, A. (2022). Learning suction graspability considering grasp quality and robot reachability for bin-picking. Frontiers in Neurorobotics, 16, 806898.

[2] Mousavian, A., Eppner, C., & Fox, D. (2019). 6-dof graspnet: Variational grasp generation for object manipulation. In Proceedings of the IEEE/CVF International Conference on Computer Vision (pp. 2901-2910).

[3] Wu, A., Guo, M., & Liu, C. K. (2022). Learning diverse and physically feasible dexterous grasps with generative model and bilevel optimization. arXiv preprint arXiv:2207.00195.

**Quality Of The Limitations Section:**

Limitations are addressed clearly

**Questions For Rebuttal:**

1.	The choice of objects is not elaborated anywhere. Are the objects seen during training?

2.	How is f_t decided in Section 4.1 “Labeling Data”? Doesn’t the choice of f_t affect whether the grasp succeeds?

3.	What is the method used to fit the function F in section 4.1 “Modeling Suction Properties”? What structure and order does F assume?

4.	More visualization should be provided for the simulation scenarios. It is difficult to judge the relevance of the simulation evaluations as is.

5.	There is no justification on how well the data labeling scheme and suction cup simulation scheme reflect real world grasp success. The simulated scenarios should be reproduced in the real world.

6.	Why does the miniscule grasp point difference in Figure 3b & 3d result in different outcomes? I suspect it is an artifact of the simulation evaluation metric. This should be clarified, preferably with real world evaluation.

7.	The remark in Sec 5.2 “Consistency Test” “our method consistently suggested grasp points within the same local region” is unclear to me. How does the size of this “local region” compare with the 2cm difference in Figure 3b & 3d?

8.	The real-world evaluation is insufficient. More scenarios and repetitions are necessary.

**Robotics Focus:**

Sufficient demonstration on hardware

**Summary Of Paper:**

The paper proposes DYNAMO-GRASP, an approach to generating suction gripper grasp points for objects without stable support, in particular in the lateral picking scenario. First, the authors implemented a simulation setup in Isaac Gym for suction gripper grasping. The simulation uses real world data to predict the forward force required to achieve a suction grasp. If the robot achieves a forward force above this value, the simulation deems it as a successful grasp. Using this method, the authors produced a dataset and trained an autoencoder that outputs an affordance map representing the estimated grasp success probability on each grasp point, the largest of which indicates the optimal grasp point. The authors evaluated their method on 50 simulated scenarios and 5 real-world scenarios. The proposed method succeeded in 98% of the simulated scenarios and 4 out of 5 real-world scenarios, which outperformed baseline methods DexNet 3.0 and suctioning on the object’s centroid.

**Summary Of Recommendation:**

Overall, the paper seems like a work in progress. It can be significantly improved with more experiments and a revised presentation. However, I am skeptical that these can be completed within the rebuttal timeframe. I do not recommend this submission for CoRL 2023.

---

### Official Review · Reviewer_GYg8 · 2023-07-19

**Confidence:** 3
**Originality:** Very Good
**Technical Quality:** Very Good
**Clarity Of Presentation:** Fair
**Impact:** 4

**Recommendation:**

Weak Accept: I recommend accepting the paper, but will not argue for my recommendation if the majority of other reviewers have a different opinion.

**Review:**

The paper discusses several interesting aspects, including the simulation environment with the analytical model of the suction cup, the grasping points sampling for training (DexNet + uniform sampling), which awards the robot kinematics, and the diverse simulated environment with domain randomization. Additionally, the proposed learning cost enables the learning of a continuous affordance map from sparse grasping score samples in the training data.

However, the paper requires some improvements in its writing, as it lacks certain essential details and is not always easy to follow. Here are some specific points to address:

- Could you provide information about the input-output dimension?
- How did you define Q, K, and V for the transformer encoder structure?
- It would be helpful if you could elaborate more on how to obtain the optimal grasp point (p, v) (line 238) from the affordance map. For example, is the largest value in the affordance map transformed into the world coordinate as p? How do you obtain v?
- You mentioned conducting a system identification process to ascertain the function F. Could you explain your 'system identification process' in more detail?
- What criteria define 'challenging cases' (line 298)?
- The phrase 'based on the following crucial criteria: (line 125)' is not a complete sentence and requires revision.
- In line 202 'robot's model', are you referring to a robot model? If so, please make it clear.
- The section titles '4.1 Simulation (line 174)' and '4.2 Robot Learning (line 230)' are rather vague and general. Consider improving them to something like '4.1 Simulation Environment and Data Collection' to enhance clarity.
- Could you explain what \hat{y}_t (line 245) represents?


**Quality Of The Limitations Section:**

Limitations are addressed clearly

**Questions For Rebuttal:**

There are two folds:
1. Clarification: Can you explain how the state transition model affects the pushing action a_t?

2. Writing: Is it possible to address the review-writing section?

**Robotics Focus:**

Sufficient demonstration on hardware

**Summary Of Paper:**

This paper focuses on suction grasp point detection. It addresses the task of determining the center of the contact ring (p) and the gripper's approach direction (v) based on a single-view depth image observation. The main contributions of the paper include: 1) Suction grasping simulation, 2) A robot learning framework for suction grasping that considers seal quality, wrench resistance, and object movement during the picking process, and 3) Real-world evaluation. To elaborate further, the grasping simulation involves the use of an analytical model for the suction cup, employing the Perimeter Springs system within the quasi-static spring system in Isaac Gym. During the inference process, the system queries grasp point candidates and outputs affordances, highlighting the optimal grasping points.

**Summary Of Recommendation:**

Suction-based robot manipulations have been studied extensively, as mentioned in the related work section of this paper. However, certain challenges persist, particularly concerning the deformable suction cup's nature, which demands vacuum sealing and pushing motion for picking, leading to the object moving while being grasped. This paper effectively addresses these challenges through sim2real techniques and demonstrates impressive real-world results. Nevertheless, there is room for improvement in terms of writing clarity.

Therefore, I recommend a weak acceptance of this paper.

---

### Official Review · Reviewer_TQDd · 2023-07-23

**Confidence:** 5
**Originality:** Very Good
**Technical Quality:** Very Good
**Clarity Of Presentation:** Excellent
**Impact:** 3

**Recommendation:**

Weak Accept: I recommend accepting the paper, but will not argue for my recommendation if the majority of other reviewers have a different opinion.

**Review:**

I find this work very timely and important for the robotic community. As explained in the introduction, state-of-the-art grasp point detection methods for suction grasping are not suitable for complex object-picking scenarios, such as object-in-shelf side-grasps, because they do not consider the movement of the object during the manipulation process. I believe that this work represents a step forward in exploring this direction. Nevertheless, I still have open questions that I think need to be addressed before considering this submission ready for publication, please see the next section.

**Quality Of The Limitations Section:**

Limitations are addressed clearly

**Questions For Rebuttal:**

1. Where does the network architecture come from? If it is adopted from previous work, this must be cited, together with the reasoning behind this decision. If it is a "novel" architecture, many more details are needed, explaining all the trade-offs and design choices that came with it.
2. Similarly, additional details are needed regarding the loss function: where does it come from, what other options were considered, why is Y_max important, and what effect does it have?
3. Additional information about the training data used: how many grasps, how many labeled points, how many images, how many GB is the total dataset, etc.
4. Additional information about the training process: how long (time and steps), what hardware was used, what hyperparameters were chosen, etc.
5. The pipeline requires segmentation masks for each object. How were these generated? Does the simulated evaluation use ground truth masks? What about the real robot experiments?
6. 50 scenarios in simulation and 5 in the real world are not so many. Increasing the number of runs would improve the statistical significance of the experiment results

**Robotics Focus:**

Sufficient demonstration on hardware

**Summary Of Paper:**

This work presents DYNAMO-GRASP, a dynamic-aware grasp point detection method, which accounts for the object dynamics during the grasp prediction process. The method is evaluated on object-in-shelf side-grasps scenarios, both in simulation as well as on a real robot, and its performance is compared to two baselines: a centroid heuristic and DexNet. The main contributions of the paper are 1) a new synthetic data collection pipeline, and 2) an experimental evaluation of the proposed method.

**Summary Of Recommendation:**

I believe that this work is addressing an important research problem and it has all the potential for a good publication. Before being able to recommend this paper, the manuscript needs to be updated with the clarifications requested above.

---

### Author Response · Authors · 2023-08-13
**More Extensive Experiments, Revised Manuscript, and Supplemental Materials**

Since the initial submission of the manuscript, we have actively improved upon this work. This has resulted in a larger dataset, a refined model, and considerably more comprehensive simulated and real-world experiments. Additionally, we have included an appendix detailing the implementation specifics and design choices for both the learning process and the simulation environment. These updates are presented in the latest revision of this paper, which is attached to each response below.

**Improvement and New Materials:**
We thank the reviewers for the time and effort invested in reviewing our work and providing such constructive suggestions. We revised the paper according to the suggestions and made the following major modifications:

- **Experiment sample size increased significantly:** Our revised simulation experiment now evaluates each method using 1,300 grasps, which is 26 times the initial sample size of 50 grasps. Similarly, the updated real-world experiment involves 375 grasps with a UR16e robot, marking a 25 times increase from our previous tests.
- **Appendix**: To enhance the clarity of the paper, we included supplemental materials that delve into the design and implementation details of the simulation, data generation, learning processes, and experiments.
- **Ablation study**: We incorporated an ablation study to demonstrate how our choice of loss function and the additional penalty term used in the labeling process contribute to the effective training of our model.

Revised Contributions:
- We describe the **challenge of complex object movement during suction grasping**, which no current state-of-the-art method adequately addresses.
- To address this challenge, we created a novel physical **simulation environment** simulates the impact of object dynamics on the success of suction grasps during the grasping process. This environment will be open-sourced and can be customized for various applicable tasks.
- Using the simulation environment, we generated a **dataset** addressing a real-world robotic challenge in an industrial warehouse context. This dataset will be made publicly available.
- We **added extensive experiments** showing that our method DYNAMO-GRASP delivers improved grasping performance with greater consistency in both simulated and real-world settings. Notably, in real-world experiments involving challenging scenarios, our method exhibits an improvement of up to 48% in success rate compared to alternative methods.

---

### Author Response · Authors · 2023-08-15
**Reminder: Discussion Phase Closing Soon**

This is a friendly reminder that the discussion phase is approaching its end, with the deadline set for tomorrow Aug 15 at 11:59 PM PT. We are willing to engage with all reviewers if possible and to provide any additional information that may be needed. Thank you for your time and thoughtful consideration of our work!

---

### Author Response · Authors · 2023-08-16
**Note on Rebuttal Responses**

Dear Area Chairs:

We sincerely appreciate and value the reviews, which have greatly assisted us in improving our manuscript. Our updated version now includes significantly expanded real-world (with 25x more samples) and simulation (with 26x more samples) experiments. We believe that this revision addresses the majority of the comments made by the reviewers. While we understand that some reviewers have not yet had the opportunity to respond to our rebuttal, we would like to kindly draw your attention to our responses, especially in the event that they are too busy to engage during part 2 of the rebuttal process.

Thank you very much for your time and consideration.

---

### Decision · Program_Chairs · 2023-08-30

**Decision:**

Accept (Poster)

**Comment:**

Following the rebuttal, the authors have actively made improvements to the paper since the initial manuscript submission. These improvements have led to a larger dataset, a better model, and more comprehensive experiments in simulated and real-world scenarios. It receives four **Weak Accept**. Therefore I would like to recommend an Acceptance.

Please revise the paper in line with the rebuttal discussion for the camera-ready submission.

Strengths:

- The work tackles a challenging and relevant problem in suction cup grasping.

- Well-motivated problem with clear relevance to robotics

- A novel, more realistic approach to simulating suction cup grasps.

- The paper is well-written and structured

Weaknesses:

- Lack of empirical real-world evaluation, especially more complex scenarios

- The collection of objects mainly comprises cylinders and box-like items, which is quite limiting. It's recommended to include more commonplace objects as seen in other research papers on grasping.